# La Tène Horse Remains from Alba Iulia CX 143 Complex: A Whole Story to Tell

**DOI:** 10.3390/ani14111624

**Published:** 2024-05-30

**Authors:** Alexandru Ion Gudea, Vitalie Bârcă, Alexandra Irimie, Cristian Olimpiu Martonos, Antonia Socaciu

**Affiliations:** 1Faculty of Veterinary Medicine Cluj-Napoca, University of Agricultural Sciences and Veterinary Medicine Cluj-Napoca, 400372 Cluj-Napoca, Romania; alexandru.gudea@usamvcluj.ro (A.I.G.); alexandra.irimie@usamvcluj.ro (A.I.); 2Institute of Archaeology and History of Arts Cluj-Napoca, Romanian Academy, 400084 Cluj-Napoca, Romania; vitalie_barca@yahoo.com; 3School of Veterinary Medicine St. Kitts & Nevis, Ross University, Basseterre P.O. Box 334, Saint Kitts and Nevis; cmartonos@rossvet.edu.kn

**Keywords:** La Tène, Iron Age, Transylvania, horse, archaeozoology, dental morphology

## Abstract

**Simple Summary:**

Archaeozoological investigation of the horse remains discovered in Alba Iulia-CX 143 complex of the La Tène period reveals the existence of a small-sized male horse (Equus caballus) of 1200–1300 mm in height with slender extremities that died at the age of 7–8 years with no evident paleopathological changes characteristic to horseback riding but showing distinctive elements of bit wear.

**Abstract:**

The present paper deals with the archaeozoological investigation carried out on a horse skeleton discovered in a Late Iron Age La Tène tomb (coded CX 143) in Alba Iulia, Romania. The paper presents all the results of the investigation, with a description of finds, adding a detailed assessment of the dentition with some interesting conclusions on the usage of a horse bit and the possible consequences of this use. The morphological features of the horse indicate a 7–8-year-old male individual, with a recalculated height of 1200–1300 mm. What is also stressed in the investigated sample is the lack of the characteristic pathological lesions typical for horseback riding but showing distinctive elements of bit wear. A comparative perspective over the few findings from the same period is provided to ensure the framing of the identified individual into the much larger historical context.

## 1. Introduction

Archaeozoology plays an important role in the reconstruction of past life. Speciality studies stress the importance of the fauna in the development and life of past human populations. A significant part of their life’s everyday activity gravitated around animal breeding, regardless of their main occupation—warriors, agriculturalists, craftsmen, etc. On the other hand, the data obtained by the study of skeletal remains of antique animals may provide significant information about the microevolution of animal populations and specific local or regional features of different species. More than that, some may indirectly provide data on human population migratory patterns and movements in different historical periods [1,2,3,4].

### 1.1. The General Historical Context

The La Tène period and culture is a period belonging to the European Iron Age Culture. The chronology of this period is from about 450 BC to the Roman Conquest and corresponds to the nowadays territories of France, Germany, Belgium, Switzerland Austria, the Czech Republic, Slovenia, Hungary and some parts of Romania—more precisely, the western part of the country (Transylvania) and western Ukraine. The term La Tène is used to characterize a site type that is similar to the homonymous site in Switzerland discovered in 1857 in the area of Lake Neuchatel [5].

The La Tène period has been divided by historians into several stages—the most used being the early, middle and late La Tène stages based mainly on the typology of metal finds, but the divisions might be still somehow debatable as scholars may refer to different points of interest in their periodization or use different systems in dividing the period (3 stages I–III or most commonly 4 stages A–D) [6,7].

Archaeological evidence indicates that, in general, the Late Iron Age chronology of the eastern Carpathian Basin went through a quite different evolution in comparison with those of Central and Western Europe due to a series of specific cultural and historical conditions [6]. The social-political and economic development of the indigenous communities from this region along the supposed traditional lines of the end of the Early Iron Age was modified in the second half of the 4th century BC by the arrival of some western groups who colonized the northern and eastern areas of the Great Hungarian Plain and most of Transylvania. First Celtic groups advanced from Transdanubia to the east, in the northern part of the Great Hungarian Plain and along the upper Tisza basin. From these areas, they moved southward along the western Carpathians and later also settled in Transylvania [6]. On the territory of Transylvania, the second Iron Age (La Tène) is divided, from cultural and historical perspectives, into two distinctive horizons. The first is the Celtic horizon (dated 350–191/175 BC) and the second is the Dacian horizon (approximately 191/175 BC to 106 AD) [8,9]. The Celtic horizon marks the early and middle stage of the second Iron Age stage in Transylvania and points towards Central and Western European cultural models. The Celts shared this territory with the indigenous populations and their interactions contributed to the hybridization of culture and material practices in such a way that the typical Central European La Tène elements were combined with the local ones [8,9]. The second horizon—the Dacian one, marks the middle and late part of the La Tène period and featured the birth of the Dacian communities in this territory and later the establishment of the Dacian Kingdom. The material culture and practices of this second horizon are different, being mainly oriented towards cultural models from the Lower Danube region and the Mediterranean area [8,9,10].

### 1.2. The Archaeological Context

A rescue archaeological study conducted by the National Museum of the Union in Alba Iulia (Romania) in partnership with Arheosib Consulting Ltd. (Sibiu, Romania) was carried out in the region of the Olympic Pool in Alba Iulia on Republicii Blvd nr 3 in March–May 2022, preceded by a first stage in 2021 that entailed non-invasive surveys that provided an image of a significant part of the whole perimeter. The land plot (more than 7500 sqm) is situated on the northern side of the city, on a high terrace, at 3.2 km northwest of Mureş and 2.1 km west of Ampoi creek, within an area that yielded multiple archaeological finds [11].

The archaeological investigations identified 216 objectives, entirely examined and excavated, dating to the Bronze and Iron II Ages, the Roman and post-Roman periods and the Modern period [11].

Among the few La Tène remains of the Olympic Poole site, we here focus on the Cx 143 context finds (Figure 1). It is a pit oriented NE–SW, sized 1.75 × 0.47 m. The maximum depth of the pit is 0.58 m from the outline level. A horse positioned with legs pulled underneath the body was discovered in the pit. Near its head, which lay to the NE, was discovered an iron bit (snaffle bit) with the mouthpiece composed of two smooth joint bars terminating in two large bit rings [11]. These bits belong to the type XIV variant A by the classification of Werner [12] or to the type II.1 [13]. These bits with large circular end rings were intensely used on a wide territory starting from the La Tène period into the Carphatian Celtic contexts but were spread also into the Lower and Middle Danube area. The time frame they can be attributed to is situated at the end of the early La Tène period (B2) and Middle La Tène (C1–C2) but they also appear later, in D La Tène complexes [12]. On the current territory of Romania, pieces (of western, Celtic origin) dated II-I BC were found in several sites [11], with an extension of usage for horse bits with round snaffle bit rings up to the first centuries AD, in Roman and even barbarian worlds. Some similar finds are noted also in the Sarmatic culture of the Pannonic Planes or late Skytic culture on the territory of Crimea [14,15].

The horse burial from this present study, along with over 180 other archaeological contexts investigated in 2018–2019 and 2022, belongs to the second period of the Iron Age (La Tène) which is richly represented in this area of the town of Alba Iulia pointing to a periodization that indicates the II-I BC timeframe, corresponding to the C1/C2 and D1 substages.

A special note on this positioning is to be made, as the pit was very narrow, giving the impression that the deposition of the horse corpse was quite unusual [16], as if it was dropped with legs down and spine up into this narrow pit, unlike another regular burial pit situated nearby, of similar dating, discovered in 2018 (complex no 401), into which another horse was placed into a much larger pit, of 2 × 1.75 m in lateral recumbency, with legs tucked under [4]. The 157 complex, of similar dating, is another horse burial into a much larger pit (3.28 × 2.76).

Horse burials are frequently found in different cultural environments of the Iron Age, accompanied or not by a variety of rituals implying also horse sacrificial offerings. Usually, horse burials are encountered in the necropolis or as isolated tombs within the settlements or as ritual depositions outside the settlements and even as ritualic complexes. Most of the time, complete horse skeletons are placed, along with various inventory pieces, most frequently harness pieces. It seems that there is no settled rule concerning body deposition, in some cases, depending on the available space, bodies are placed either on their right or left side with or without the legs tucked under. This complete carcass deposition suggests that most of the time these were sacrificial rituals, but it does not exclude other simpler reasons such as economic carcass depositions in certain conditions [16,17,18].

Among the inventory pieces of these horse burials, many bronze finds related to harness elements and decorations have been found. The presence of such finds indicates a possible presence of barbarian mercenaries originating from the northern area of the Black Sea or the fact that the horse itself might have been brought into this space by the local warriors that were located once briefly in the area of the Black Sea. The existence of such artefacts is a clear indication of palpable cultural links with the northern and northwestern Pontic spaces [14].

## 2. Materials and Methods

The bony material harvested from the excavation site was presented to the Anatomy Lab of the Faculty of Veterinary Medicine Cluj-Napoca Romania. The material was cleared, washed and gently cleaned from all debris in the Anatomy lab.

The pieces of the skeleton were significantly affected by disarticulation and dispersal processes, but diagenetic fossilization and later mechanical alterations (including exposure to soil moisture and acidity) made the investigation more difficult than usual, as the bones were very frail and prone to further fragmentation [19].

The bones were left to dry naturally for a certain amount of time, after which a general assessment was made to establish the optimal procedures for the identification. As the bones were intensely affected by the exposure to physical agents (soil acidity and humidity), most of the pieces called for a consolidation procedure. A construction product based on PVA (polyvinyl acetate) was used in the primary approach in an attempt to preserve and consolidate the frail elements and to connect the extremities of the long bone fragments [20]. Another drying phase followed, and a proper analysis was initiated by evaluating the degree of fragmentation and the potential of rebuilding the separated parts of the bones.

After the manual reconstruction of some of the long bones, mainly (extremities and shafts), the unidentifiable pieces were counted and separated from the identifiable material. The specific morphological identification was made [3,21,22,23,24,25,26,27] and the osteometric evaluation was performed based on standard measurements [28]. Evaluation of the epiphyseal stages and dental data was made based on the standard archaeozoological protocols [21,27,29].

## 3. Results

### 3.1. Representation of the Skeletal Fragments

The sample consists of remnants originating from one single individual, distributed as notated in the following table (Table 1).

### 3.2. The Appendicular Skeleton

#### 3.2.1. The Scapula

The most relevant fragments identified were the ones from the articular angle.

Reconstruction work permitted the restoration supraglenoidian of the tuberosity. (Figure 2). This is an indicator of an age over 10–12 months [3,21]. The right scapula displayed a pathological mark, most probably a consequence of a periosteal reaction around the scapular neck, with osteophytes typical for such a reaction. A limited number of measurements (Table 2) were possible due to the high fragmentation of the piece (Table 3).

#### 3.2.2. The Humerus

For the stylopodium of the forelimb (Figure 3), we reconstructed most of the left piece, while the right one remained incomplete. This allowed an assessment of the fusion stage -s, an indicator of an individual older than 42 months [3,21]. The only length measurement (although approximate) allows a theoretical recalculation of height of 1193 mm [1,3,30].

#### 3.2.3. The Radius and Ulna

The reconstruction of radial fragments offered us the possibility to use a quite complete set of measurements (Table 4), mainly on the radial fragments. Fusion data indicate an individual older than 42 months [1,3]. Metric data allow the recalculation of an individual around 1400 mm in height (1419–1393 mm) [1,31,32].

#### 3.2.4. Metacarpals

Based on the reconstruction performed (Figure 4), an almost complete left main metacarpal was obtained, while the right one was presented just as a proximal fragment. Bones are ossified, an indicator of an individual over 15 months of age [1,3]. Based on the available metrical data (Table 5) for the left piece, a recalculated height of 1230 mm was obtained [1,3]. The slenderness index (calculated as the proportion of the greatest length (1) and the distal breadth (6)) is 14.8, a number that includes our individual in the semi-slender group, in accordance with the Brauner scale [3].

#### 3.2.5. The Coxal Bone

Eight fragments of coxal bone were identified. Due to a high degree of fragmentation, the assessment was conducted only on some iliac body fragments (2), one alar part and three periacetabular ones. The only conclusion we can draw based on the ossification stage of the area of the iliac crest is the fact that the individual was older than 4.5–5 years [1,3].

#### 3.2.6. The Femur

The identified pieces (Figure 5) and the reconstruction did not provide sufficient metrical data (except for measurements of the minimum breadth of the shaft).

Ossification data indicate the origin from an individual older than 42 months [1,3].

#### 3.2.7. Tibia

Two relatively complete bones (Figure 6) were identified. Based on the fusion data [3,21,33], we concluded that the fragments originate from an individual older than 42 months.

The complete length of the bones (Table 6) allowed us an estimate of the shoulder height of 1294 and 1308 mm [1,3].

#### 3.2.8. Tarsals

Only two fragmented calcaneal bones were clearly identified. The only assessment made from these bones was the fact that the tuber was ossified, an indicator of age over 36 months [3]. No metric data could be collected. The same goes for the talus fragment identified.

#### 3.2.9. Metatarsals

The identified fragments did not allow measurements, except for some unimportant data from the shaft area (Table 7). Few conclusions can be drawn from these elements, the main one being an estimated age of over 15 months [1].

#### 3.2.10. Phalanges

In this group, we included all phalanges, regardless of their origin in the fore- or hindlimb. All fragments are ossified, an indicator of an individual older than 10–15 months [3]. Collected metrical data are presented in Table 8.

### 3.3. The Axial Skeleton

#### 3.3.1. The Mandible

For this category, 25 fragments were identifiable. As the pieces were so small and frail, a reconstruction of the whole mandibular piece was not possible; thus, no morphological and metrical evaluations were available. About 5 fragments were more or less identifiable as parts of the mandibular curved part while some others were presented as highly fragmented alveolar walls. No evaluation could be made on the diastemal part of the mandibular body.

#### 3.3.2. Ribs

Of the total identifiable pieces (134), only a few were proximal articular fragments, in a bad state of preservation. No data can be gathered on their basis.

#### 3.3.3. Vertebrae

Most of the fragments were vertebral bodies. The fusion of the bodies was a clear indicator of an age over 4.5–5 years [3]. About 14 fragments were represented by vertebral arches and spinous (ossified) processes of thoracic vertebrae while only 7 fragments were listed as cervical vertebral fragments. No pathological or subpathological elements were visible on the assessed fragments.

#### 3.3.4. Dentition

Despite the heavy fragmentation of the entire sample presented, we could recover and identify all the dental series elements. This permitted an exact evaluation of the age of the individual and some investigations that facilitated the species’ precise identification.

For incisors, the anatomical arrangements and identification of the complete series (101–103, 201–203 the upper row and 301–303, 401–403 as the lower row) permitted a whole range of evaluations.

The dentition displays the difference in wear stages between the upper and (Figure 7) the lower incisive (Figure 8) series. There is a higher degree of wear seen in the lower series than in the upper one, a fact that, to a certain extent, is quite normal [34,35]. Moreover, the attrition pattern in the lower incisors shows abnormal wear, with the intermediate and lateral incisors (corner) worn to a much higher degree than the central one. Age estimation based on the central incisor suggests a minimum age of 6–7 years (as the cup is gone), while this abnormal wear on the next ones could have been an indicator of a higher age [36,37,38]. Such unusual wear patterns, as mentioned in modern veterinary sources, appear as a consequence of cribbing or some unusual stereotypical chewing patterns (maybe wood chewing) [39,40,41,42,43], thus not excluding a similar situation for the studied individual as well.

For the upper incisors, the wear is different than expected. There is a much smaller degree of wear; all of the upper incisors show the existence of the cup (with a normal pattern of wear from the central ones to the lateral ones), indicating a quite similar age to the one assessed based on the lower incisive series. One extra element—the apparent dovetail notch on the upper lateral incisors—indicates an individual 7–8 years old [1,34,35,36,37,38].

All 4 canines (Figure 9) were identified, showing a certain degree of wear. Their existence is an indicator of a male individual, with an age above the limit of 4.5 years [3,21].

For the molar series, an overview of the entire set indicates an age over 4 years [21]. The overall wear is not advanced and the molars show no unusual wear patterns (Figure 10 and Figure 11).

Our investigation focused on one of the most relevant anatomic components of dentition, so we could rule out the possible misdiagnosis of the species.

One of the typical elements of differentiation used as differential criteria is the enamel pattern on the grinding teeth, in particular the asymmetry between the metaconid and metastylid on the lower molar series (Figure 12). The lingual direction of the line between these enamel folds forms the so-called lingual valley, which is U-shaped in our case [44]. All these elements are a good indicator of the horse as a species. The third element which may also serve as the differential criterion is represented by the buccal fold or vestibular groove of enamel that penetrates from the space between the hypoconid and the protoconid (the preflexid–postflexid interstice), which is not as deep as the one typical for donkeys.

More than that, the typical folding called Pli caballinid is easily identifiable on 2nd premolars and molars [25,26,44,45].

For the upper dental series (Figure 13), it is acknowledged in several literature sources [44] that the spaces/arches between the mesostyle and the parastyle are quite deep and well-profiled, as the styles are thick. The (lingual) protocone has an elongated shape, with a larger posterior expansion. This is visible on the specimen from our studied specimen, confirming again our initial identification.

Sources cite even the calculation of a protocone index (expressed as a ratio between the length of the protocone and the occlusal length of the tooth) as a discriminatory element for horses, *E. hydruntinus*, mules and *E. Ferus* [44] but, as comparative data are lacking at the present moment, we appeal to the standard of the elongated shape for horses and the short, narrow shape for donkeys [44]. The pli cabalin is usually identifiable, in contrast to the unidentifiable status of this fold in donkeys (Figure 13).

The other interesting focal point [41,42,43,46,47,48] was the aspects that resulted from the first lower premolar’s (306 and 406) assessment. As mentioned in many literature sources, this piece of dentition presents a special wear pattern in the case of the bit usage [46,49].

As the conductor pulls the metallic bit (which normally rests on the tongue and gums in the diastemal area (the bars)) to one side or another, it forces the animal to turn its head in the desired direction. To avoid pain, the animal uses its tongue to lift the bit and bring it towards the occlusal surface of the second premolar. This is the area that is usually affected by a heavier wear than the rest of the occlusal molar surfaces, causing mainly a flattening of the first cusp (protoconulid). Literature sources even suggest a method of measuring the difference in height between the line of metaconid and metastylid eminences and the level of the protoconulid on the mesial margin of the tooth to assess the impact of bit wear on these teeth [46]. As visible in the image (Figure 14), our pieces show evident wear on the anterior part of the occlusal surface. Another indication of the bit wear is the exposure of the enamel and dentine on this anterior (mesial) margin of the premolar, with this (almost) sharp anterior edge progressively getting worn with the enamel and then the dentine exposure. This strip increases in width as the wear progresses [47]. The earlier-mentioned authors also suggest a method of evaluation based on a ratio of the height and width of the exposure to assess the intensity of the wear. As we can see from the figure (Figure 14), the investigated specimens present a discreet but visible strip of dentine exposure in both premolars, a good indicator of bit usage and wear.

## 4. Discussion

The sum of data allows us to conclude that the studied skeleton from pit Cx143 from Alba Iulia shows some peculiar features. It begins with an initial burial which is different from most of the ritual burials documented in the area, where most of the animals were deposited lying on the side, sometimes with the distal part of the limbs tucked under, over flexed or even disarticulated so the body fit into the much wider burial pit. The presence of the bit is, following archaeologists, solid proof of the use of the animal as a riding or draught animal, not excluding being used with a chariot as well.

The morphological data, as recorded so far, do not indicate a typical riding or draught animal, since pathological evidence is missing [50,51,52], but show another, most probably particular, type of pathology associated with the osteoarticular system at the level of the left shoulder joint that might indicate a conformational fault or a traumatic cause (this pathological specimen is under a complex set of investigations in the Pathology and Radiology Department and will be subject of another more in-depth specialized investigation).

To have a larger perspective on the characteristics of the horse population in the La Tène period, the morphological characteristics of horses from several comparable sites were analyzed and illustrated (Figure 15).

We compared the metric data from an earlier study on a horse from the same archaeological objective, published in 2020 [4], the Zimnicea Necropole [53,54,55], another one from the same county [56], some horse remains from a La Tène site in Savârsin, Arad county [57], horse skeletons from the current territory of Hungary (Scythian cemetery) at Szentes-Vekerzug and a so-called Venetian horse skeleton from the cemetery complex of Sopron-Krautacker [58]. We used most of the initial published metrical data to try to recalculate the height of the animals and the indices that help us in the classification and characterization of the morphological features of the animals [59,60,61]. Recalculation of height data was carried out following the method suggested by Bartosiewicz [62], which has been used in all recalculations (whenever available data) in all cases of studied complete horse skeletons from Romania or elsewhere [3,4,62,63,64,65,66].

As we can conclude, the studied individual was a mature male of 1230 mm height (as the above-mentioned combined formula indicates), with a value of the slenderness index of metacarpals that places our individual at the limit between slender and semi-slender appendages.

As visible in the above graph (Figure 16), the individual from Alba-CX 143, closely situated to the one identified in Savârșin (Arad), stands out in terms of gracility and height (the largest left registry dot), being one of the smallest in size among the compared ones. It is even very different from the horse identified in its immediate vicinity (Alba Olimpic Pool), which is much taller, but similar as far as the gracility of the metacarpals is concerned. The most robust individuals were the ones in Zimnicea, but their indices barely fall into the “medium” category. What seems to be evident is the fact that, if we take into consideration the so-called arbitrary limit (set to 1400 mm height), several individuals fall into that more compact group called “elite horses”, and most of them are in the slender/semi-slender group (see the shadowed ellipse on the graph), except maybe for some individuals from Szentes Vekerzug, while the group of “ordinary horses” show a much ampler array of features, with different heights and a general tendency towards very slim extremities (including the present investigated individual from Alba CX 143).

Many other sources mention the European horses as originating from two distinctive sources or belonging to two groups—the eastern group, which refers to large, strong(er) horses, and the western group, which encompasses smaller and more slender individuals [58,67]. Such a division—although leaving aside some other questions in terms of territorial separation—might indicate some degree of overlap between the aforementioned categories (elite and ordinary), despite the difference in the mentioned overall height limits. These divisions might have a significant amount of overlap in terms of the threshold value of the height and even slenderness indices. Something that also seems to plead in favor of the overlap of the two categorizations is the separation of the morphological features of the groups—one, more uniform, slightly larger as far as height is concerned (the eastern group, more probably the elite group) and the other more diverse in conformations, with smaller values for height (the western group—ordinary, local horses, such as the one that was identified in this study).

## 5. Conclusions

The discovered horse of the CX 143 pit from Olympic Pool in Alba Iulia was buried in a particular way, slightly different from the usual burials documented in La Tène sites. Our research identified a male individual of 7–8 years, with a recalculated shoulder height of 1200–1300 mm (1230 mm as average value) with slender extremities. The identified features of the dentition pinpoint the use of the bit (which has been discovered on site as well) and still leave open the question of the main usage for this animal, as specific lesions associated with horseback riding or draught animal-associated bony features were not demonstrated. A very general framing into the historical context (as far as horse morphology is concerned) indicates the placement of our individual into the group of “ordinary horses” or the western group.

## Figures and Tables

**Figure 1 animals-14-01624-f001:**
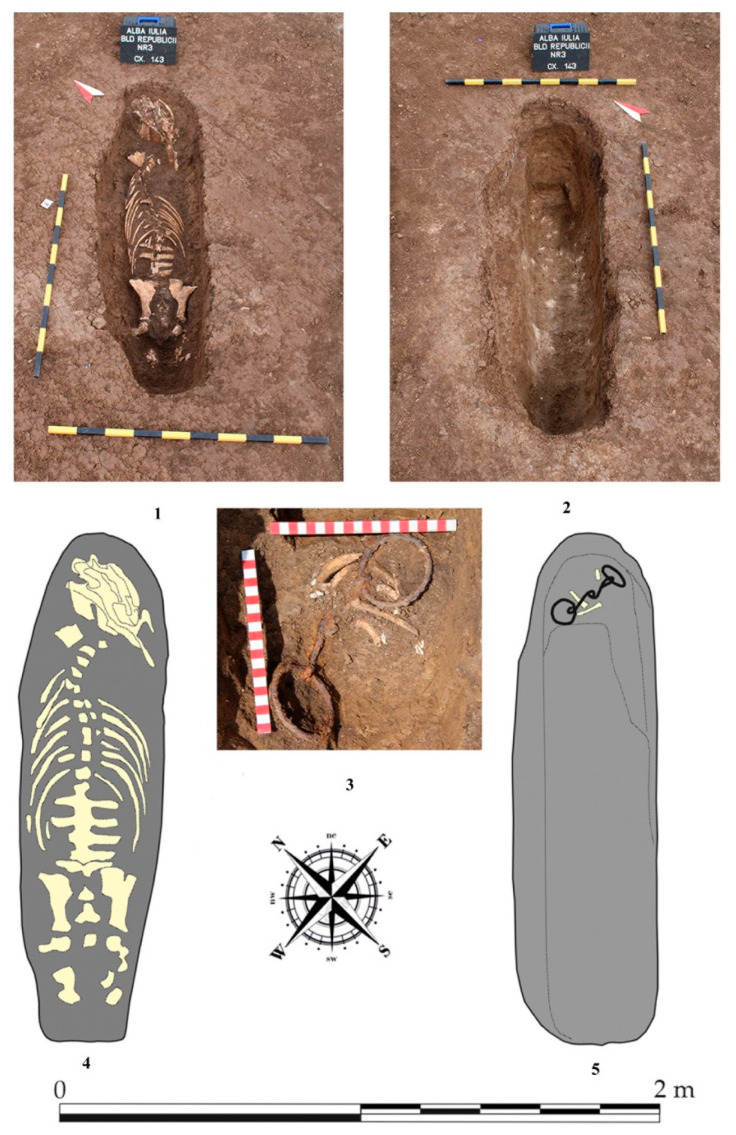
The CX 143 pit with the skeleton (**1**) and after the removal of the skeletal parts (**2**). Graphical representation of the skeletal components inside the grave (**4**) and the discovered bit (**3**,**5**).

**Figure 2 animals-14-01624-f002:**
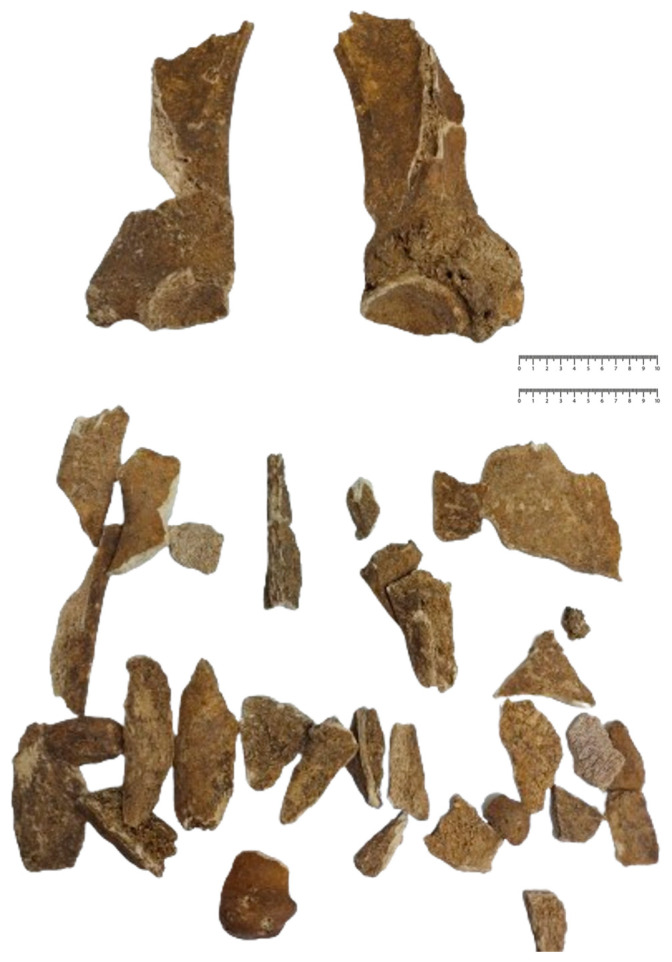
The identified scapular fragments and partial reconstruction of the bones.

**Figure 3 animals-14-01624-f003:**
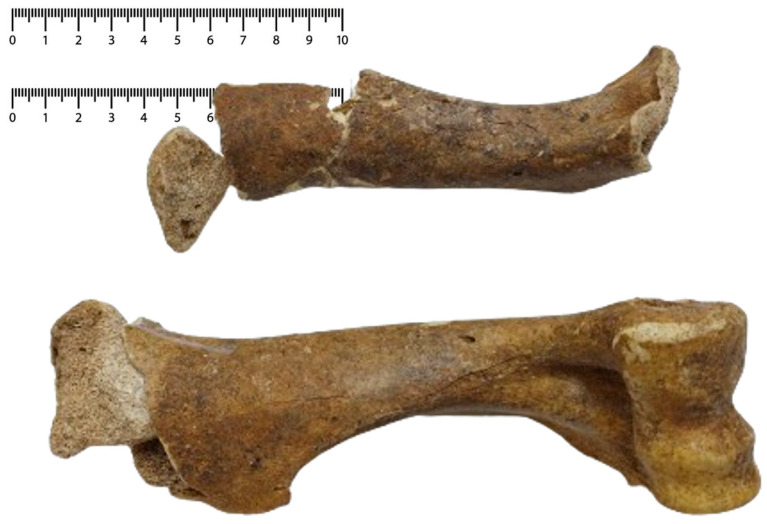
The identified humeral fragments and partial reconstruction of the bones.

**Figure 4 animals-14-01624-f004:**
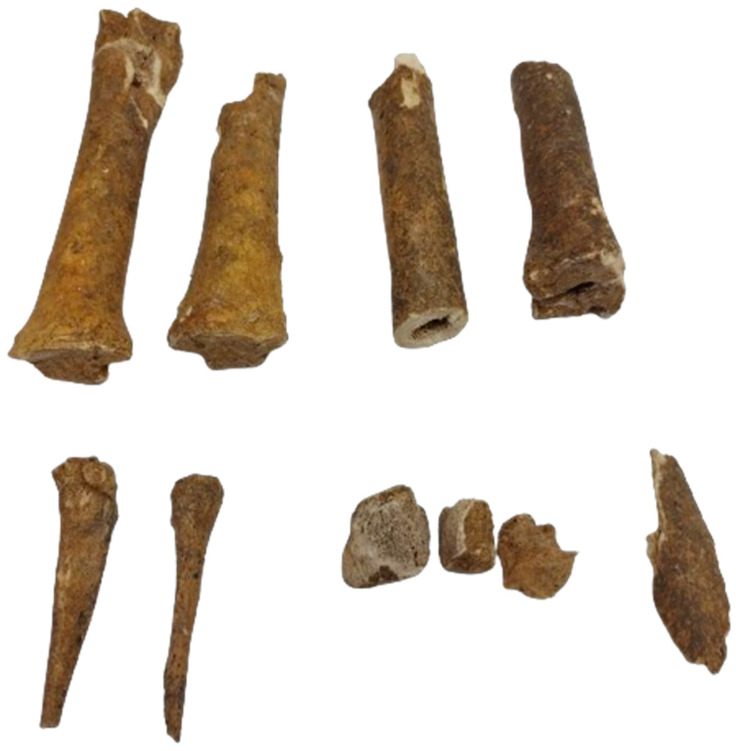
The identified metacarpian fragments and partial reconstruction of the bones.

**Figure 5 animals-14-01624-f005:**
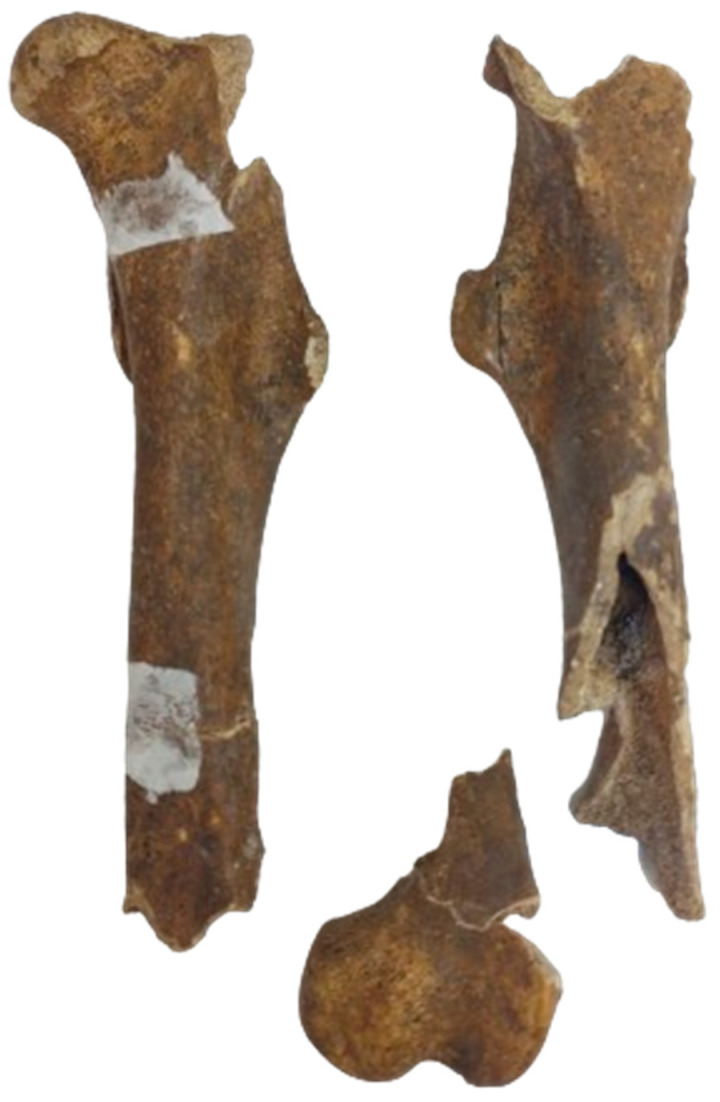
The identified femoral fragments and partial reconstruction of the bones.

**Figure 6 animals-14-01624-f006:**
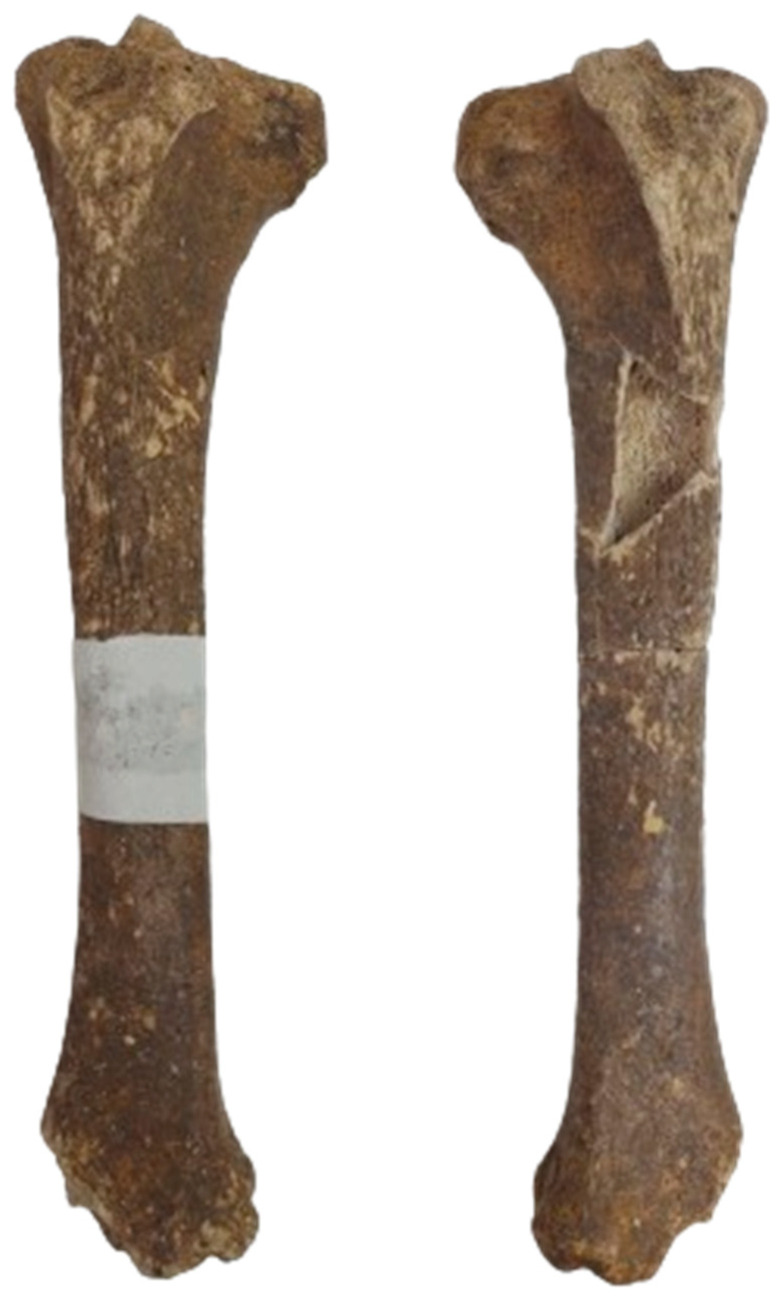
The identified tibial fragments and partial reconstruction of the bones.

**Figure 7 animals-14-01624-f007:**
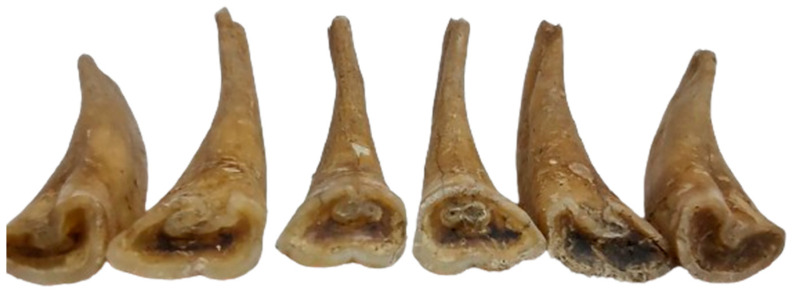
The incisors of the lower arch (301–303; 401–403).

**Figure 8 animals-14-01624-f008:**
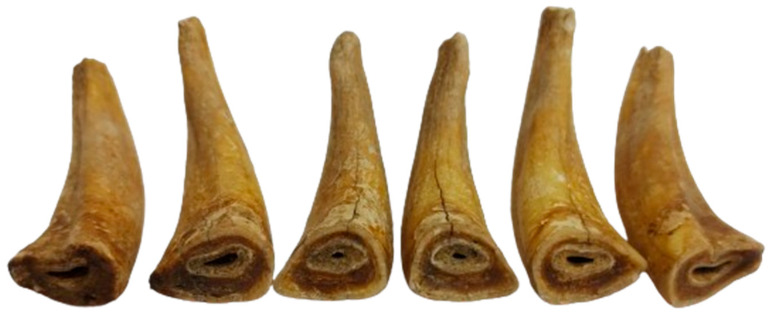
The incisors of the upper arch (101–103; 201–203).

**Figure 9 animals-14-01624-f009:**
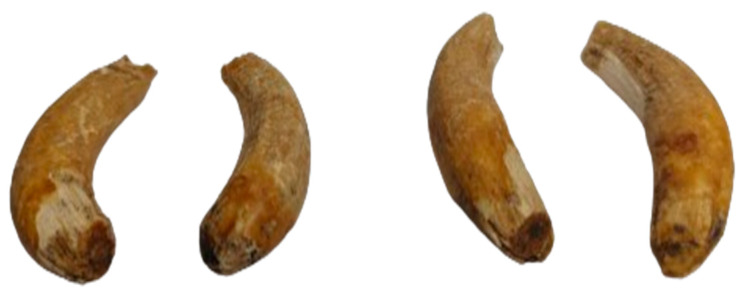
The canines.

**Figure 10 animals-14-01624-f010:**
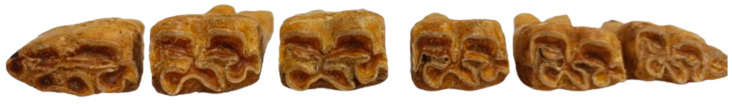
The upper cheek teeth series.

**Figure 11 animals-14-01624-f011:**
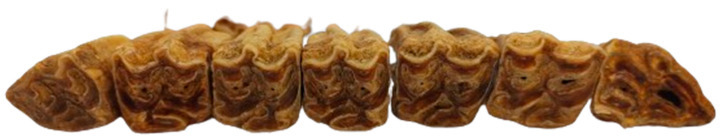
The lower cheek teeth series.

**Figure 12 animals-14-01624-f012:**
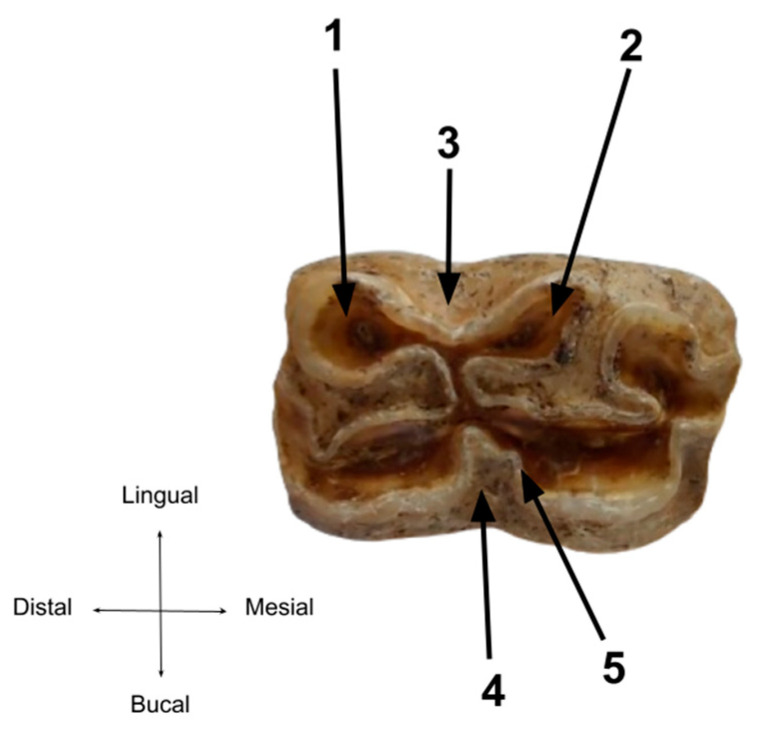
Detailed perspective on the enamel patterns on a 2nd lower molar. (1) Metaconid, (2) metastylid, (3) the lingual valey/groove, (4) the vestibular groove, (5) Pli cabaliinid.

**Figure 13 animals-14-01624-f013:**
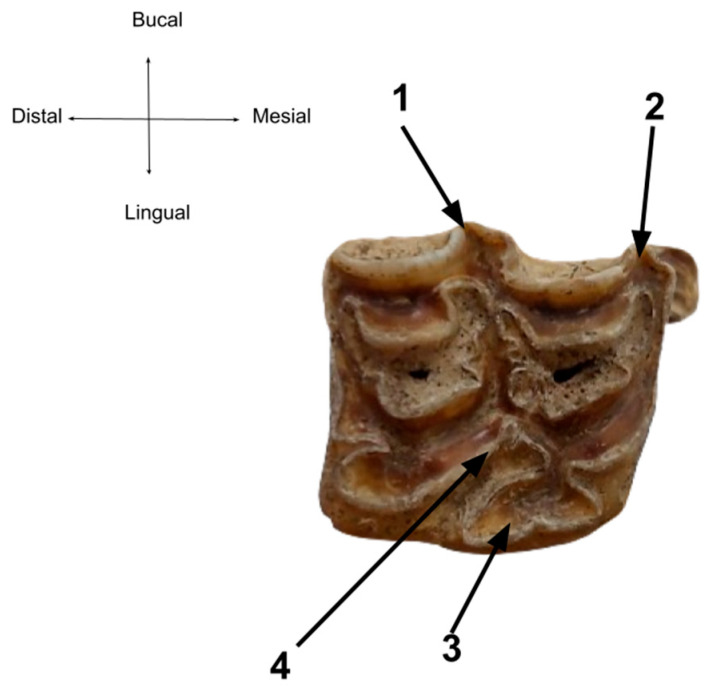
Detailed perspective on the enamel patterns on a 2nd upper molar. (1) Mesostyle, (2) parastyle, (3) protocone, (4) Pli caballin.

**Figure 14 animals-14-01624-f014:**
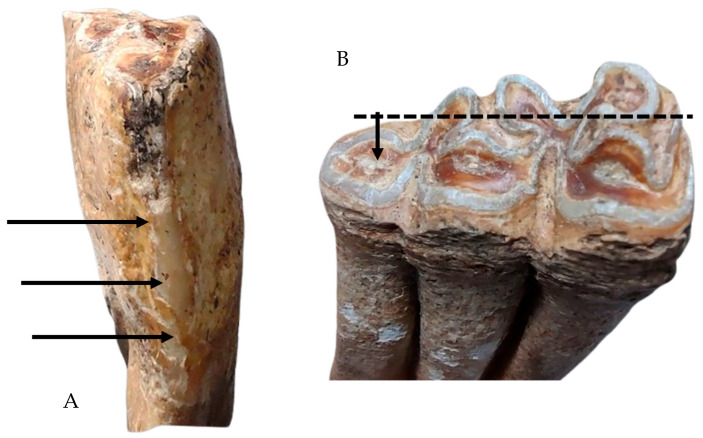
Detailed perspective on the mesial (**A**) and occlusal surfaces (**B**) of the lower premolar showing typical traces of bit wear.

**Figure 15 animals-14-01624-f015:**
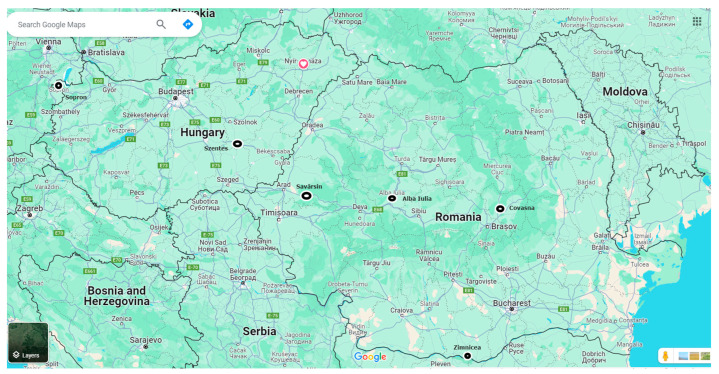
Location of La Tène sites used as references for horse materials (black dots).

**Figure 16 animals-14-01624-f016:**
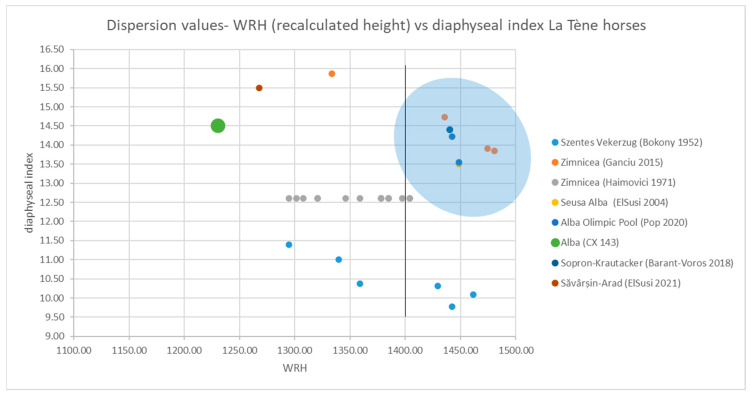
Metrical values for metapodials of La Tène sites (WRH height vs. diaphyseal index) [4,53,54,55,56,57,58].

**Table 1 animals-14-01624-t001:** Numeric representation of the identified skeletal fragments.

Skeletal Element	R	L	Unclear
Scapula	1	1	32
Humerus	1	1	4 Proximal fragments
Radius and ulna	1 radius + 1 olecranon	1 radius + 1 olecranon	
Metacarpals (III)	1	1	
Carpals			3
Coxal			8
Femur	1	1	
Tibia and fibula	1	1	
Tarsals	1 calcaneus	1 calcaneus	1 talus, 1 centrotarsal
Metatarsals (III)	1	1	2 shaft fragments
Phalanges			2 PH 1, 1 PH 2, 1 proximal part PH1
Sesamoid bones			2 sesamoid bones and 2 patellar fragments
Rudimentary metacarpals and metatarsals			4
Vertebrae			84
Ribs			134
Dentition			12 incisors, 12 PM, 12 M, 4 C
Condylar fragments-long bone			15
Bone shaft fragments-long bone			29
Mandibular fragments			25
UNIDENTIFIED			320

**Table 2 animals-14-01624-t002:** The metrical data for the scapular fragments.

Scapula-Metrical Data	R	L
smallest breadth of the neck (4)	55	
anteroposterior diameter of the articular process (5)	76	77
anteroposterior diameter of the glenoid cavity (6)	50	Approx. 51

**Table 3 animals-14-01624-t003:** The metrical data for the humeral fragments.

Humerus-Metrical Data	L	R
maximal length at the level of the head (3)	245	
minimum breadth of the shaft (6)	31.2	32
breadth of the trochlea (8)	66	

**Table 4 animals-14-01624-t004:** The metrical data for the radial fragments.

Radius-Metrical Data	L	R
greatest length (11)	350	
physiologic length (12)	327	Approx. 321
lateral length (13)	320	324
transversal diameter of the proximal articular facet (15)	76	
breadth of the proximal extremity (14)	66	
breadth of the shaft (16)	33	33
breadth of the distal extremity (18)	Approx. 56	54
transversal diameter of the proximal articular facet (19)	63	65

**Table 5 animals-14-01624-t005:** The metrical data for the metacarpian fragments.

Metacarpals III-Metrical Data	L	R
external length (3)	192	
breadth of the proximal extremity (4)	43.7	43.9
transversal diameter of the proximal extremity (5)	28.5	26.8
breadth of the shaft (6)	28.8	29.5
anteroposterior diameter of the shaft (8)	20.2	
breadth of the proximal extremity (9)	Approx. 42	

**Table 6 animals-14-01624-t006:** The metrical data for the tibia.

Tibia Metrical Data	L	R
greatest length (1)	322	320
lateral length (2)	300	297
breadth of the proximal extremity (3)	81	82
breadth of the shaft (4)	34	34
minimal perimeter of the shaft (5)		
breadth of the distal extremity (6)	Approx. 58	Approx. 56
anteroposterior diameter of the distal extremity (7)	35	33

**Table 7 animals-14-01624-t007:** The metrical data for metatarsals.

Metatarsals-Metrical Data	L	R
breadth of the proximal extremity (4)	40.2	
transversal diameter of the proximal extremity (5)	38.1	
breadth of the shaft (6)	26.4	26.7
anteroposterior diameter of the shaft (8)	26.2	25.4

**Table 8 animals-14-01624-t008:** The metrical data for phalanges.

Phalanges-Metrical Data	PH1	PH1	PH2
greatest length (1)	70.5	71	35
breadth of the proximal extremity (2)	45		
anteroposterior diameter of the proximal shaft (4)	30	32	
breadth of the shaft (5)	28	29	36
breadth of the distal extremity (6)	36	38	
transversal diameter of the distal articular facet (7)	35	34	

## Data Availability

The original contributions presented in the study are included in the article material, further inquiries can be directed to the corresponding author/s.

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
