# Peer review of "La Tène Horse Remains from Alba Iulia CX 143 Complex: A Whole Story to Tell"

_animals, 2024, doi:10.3390/ani14111624_

Round 1

Reviewer 1 Report

Comments and Suggestions for Authors

I think that the manuscript is a good contribution about specific remains of a horse which allow an interesting study for the region and time period. In my view, the authors should greatly improve the figures, specially those showing bones, to make them publication ready. In particular, it would be preferable to remove all te background from the pictures, have an uniform color as the new background, and provide a scale bar with a black line in a corner. Furthermore, the English should be checked by a native speaker to ensure that no errors are present in the manuscript. I think that those improvements will make the presentation of the study more professional and ready to publish.

Comments on the Quality of English Language

The English needs to be improved as some grammatical and punctuation errors are found along the manuscript.

Author Response

Reviewer 1

Dear reviewer,

Thank you very much for your thoughts and comments on our material. We have treated all notes with responsibility and several changes were made to our material. Hope these changes and notes have met your expectations! Below is a point-by-point response to your comments

I think that the manuscript is a good contribution about specific remains of a horse which allow an interesting study for the region and time period.

Appreciate!

In my view, the authors should greatly improve the figures, especially those showing bones, to make them publication ready. In particular, it would be preferable to remove all te background from the pictures, have an uniform color as the new background, and provide a scale bar with a black line in a corner.

Made the removal of all backgrounds

Furthermore, the English should be checked by a native speaker to ensure that no errors are present in the manuscript. I think that those improvements will make the presentation of the study more professional and ready to publish.

Text has been once again checked by a professional

Reviewer 2 Report

Comments and Suggestions for Authors

This paper offers a comprehensive examination of domestic horse remains unearthed at the Iron Age archaeological site Alba Iulia CX 143. Through meticulous description, analysis, and classification of habitus types, the authors provide a thorough account of key characteristics of the horses. This includes considerations such as the stage of ontogenetic development of the skeleton, dentition wearing, as well as the identification of pathological formations and bit marks, shedding light on the potential uses of these animals.

Structured with clarity, the article employs an academic language that enhances understanding. The figures presented are of high quality, effectively complementing the material description and discussion. Furthermore, the bibliography is comprehensive, encompassing both classical and recent publications of significance.

I have no objections to the scientific content of the paper; however, I do have some minor suggestions:

- There is inconsistency in the spelling of "La Tène," with both "La Tène" and "La Tene" being used in different instances (e.g., page 3, line 76 vs. page 2, line 42). It is necessary to standardize the spelling throughout.

- In the "Materials and Methods" chapter, according to which authors (with references) were measured skeletal remains, and provide the explications to applied measurements. Readers without access to your sources may find it difficult to understand which measurements are presented in the tables.

- On page 13, line 273, please italicize the species name "E. hydrutinus."

- Figure 16 includes abbreviations (Gl and WRH) that are not explained in the text. Please add explanations for these abbreviations in the "Materials and Methods" chapter. Additionally, there is confusion in the figure caption where it states "Gl vs diaphyseal index," while the plot shows 'WRH' and 'diaphysal index.' Please ensure consistency between the title and elements of the diagram

Once these modifications are addressed, I will be happy to recommend the publication of the paper titled "LaTene horse remains from Alba Iulia CX 143 complex. A whole story to tell" in the MDPI journal "Animals”.

Author Response

Dear reviewer,

Thank you very much for your thoughts and comments on our material. We have treated all notes with responsibility and several changes were made to our material. Hope these changes and notes have met your expectations! Below is a point-by-point response to your comments

This paper offers a comprehensive examination of domestic horse remains unearthed at the Iron Age archaeological site Alba Iulia CX 143. Through meticulous description, analysis, and classification of habitus types, the authors provide a thorough account of key characteristics of the horses. This includes considerations such as the stage of ontogenetic development of the skeleton, dentition wearing, as well as the identification of pathological formations and bit marks, shedding light on the potential uses of these animals.

Structured with clarity, the article employs an academic language that enhances understanding. The figures presented are of high quality, effectively complementing the material description and discussion. Furthermore, the bibliography is comprehensive, encompassing both classical and recent publications of significance.

Thank you very much

I have no objections to the scientific content of the paper; however, I do have some minor suggestions:

- There is inconsistency in the spelling of "La Tène," with both "La Tène" and "La Tene" being used in different instances (e.g., page 3, line 76 vs. page 2, line 42). It is necessary to standardize the spelling throughout.

Made the corrections. Using “La Tène” all over the text

- In the "Materials and Methods" chapter, according to which authors (with references) were measured skeletal remains, and provide the explications to applied measurements. Readers without access to your sources may find it difficult to understand which measurements are presented in the tables.

Modified all tables, inserting the name of the measurement, keeping the nomination (as numerals in between brackets) as in Dessee

- On page 13, line 273, please italicize the species name "E. hydrutinus."

corrected

- Figure 16 includes abbreviations (Gl and WRH) that are not explained in the text. Please add explanations for these abbreviations in the "Materials and Methods" chapter. Additionally, there is confusion in the figure caption where it states "Gl vs diaphyseal index," while the plot shows 'WRH' and 'diaphysal index.' Please ensure consistency between the title and elements of the diagram

Corrected, replaced the graph and changed the title

Once these modifications are addressed, I will be happy to recommend the publication of the paper titled "LaTene horse remains from Alba Iulia CX 143 complex. A whole story to tell" in the MDPI journal "Animals”.

Thanks!

Reviewer 3 Report

Comments and Suggestions for Authors

I liked this study overall and this is a case where the more information we have about the nature of La Tene horses and burial practices the better. It may not be a groundbreaking study but it is one of those articles that do advance the field of archaeology by providing further information and detail about La Tene animal husbandry. I also thought the solid methodology applied and careful work done in preparing, describing, photographing, and measuring the bones was evident.

I have attached a file with some alterations that should be made. Some of them are minor improvements of the English or clarifications to make. There are, however, some more significant issues that should be addressed as detailed below.

1. The introduction should include some material about what is known of the Celtic use of horses in the Tene period and especially about the nature of and reasons behind the practice of burying horses. Relevant sources should be cited here.

2. The tables use standardized measurements and merely refer to them by number while citing references. This is sometimes done but I would prefer also giving a brief description of the measurement in addition to the number so readers do not have to look this up if unknown. So, for example, a measurement can read greatest [or total] length (1), proximal width (5), etc.

3. Some of the material included in the results should be moved to the discussion [as indicated in attached document]. And the discussion should be expanded to include thoughts on reasons why this horse might have been buried in a way different from other horses, as well as how this horse might have been used given the osteological and bit findings.

4. Any gross morphological differences from other La Tene horses should be noted, or if none this should then be stated. So besides measurements and gracility index were there any significant differences in dentition, postcranial tuberosities, etc. 

5. It should be noted if the bit found here resembles or is different from in any way other La Tene bits.

Comments on the Quality of English Language

The English writing is good overall but on occasion the wording can be improved or style improved (see comments on attached file).

Author Response

Dear reviewer,

Thank you very much for your thoughts and comments on our material. We have treated all notes with responsibility and several changes were made to our material. Hope these changes and notes have met your expectations! Below is a point-by-point response to your comments

liked this study overall and this is a case where the more information we have about the nature of La Tene horses and burial practices the better. It may not be a groundbreaking study but it is one of those articles that do advance the field of archaeology by providing further information and detail about La Tene animal husbandry. I also thought the solid methodology applied and careful work done in preparing, describing, photographing, and measuring the bones was evident.

Thank you!

I have attached a file with some alterations that should be made. Some of them are minor improvements of the English or clarifications to make. There are, however, some more significant issues that should be addressed as detailed below.

Went through the pdf file provided and made the corrections accordingly! Thanks for your kindness

  1. The introduction should include some material about what is known of the Celtic use of horses in the Tene period and especially about the nature of and reasons behind the practice of burying horses. Relevant sources should be cited here.

A small section approaching this subject has been inserted, As the topic is quite vast, this is not a very comprehensive part, but sends the reader to a series of main trends and some references

  1. The tables use standardized measurements and merely refer to them by number while citing references. This is sometimes done but I would prefer also giving a brief description of the measurement in addition to the number so readers do not have to look this up if unknown. So, for example, a measurement can read greatest [or total] length (1), proximal width (5), etc.

Made all these changes- used the nomination alongside the Dessee numbers

  1. Some of the material included in the results should be moved to the discussion [as indicated in attached document]. And the discussion should be expanded to include thoughts on reasons why this horse might have been buried in a way different from other horses, as well as how this horse might have been used given the osteological and bit findings.

Done that. Thanks also for the other observation regarding the conclusions!

  1. Any gross morphological differences from other La Tene horses should be noted, or if none this should then be stated. So besides measurements and gracility index were there any significant differences in dentition, postcranial tuberosities, etc.

This is quite a delicate thing- as the previous analysys (done also by me and my colleagues on another horse skeleton) was made long time ago (2008?), I did not record any “variabilities”..and unfortunately the present material is highly fragmented and degraded. The other cited sources stated nothing about anatomical variations…so I left everything like that. One of the plans is to attempt at least a dental differential approach for the La Tene horses (another one has been harvested this year from the same area that,so most probably another comparative study may be prepared in the future).

  1. It should be noted if the bit found here resembles or is different from in any way other La Tene bits.

Several lines dealing with the typology of the bit were inserted into the introductory part- the archaeological context